# Overexpression of Type 1 and 2 Diacylglycerol Acyltransferase Genes (*JcDGAT1* and *JcDGAT2*) Enhances Oil Production in the Woody Perennial Biofuel Plant *Jatropha curcas*

**DOI:** 10.3390/plants10040699

**Published:** 2021-04-05

**Authors:** Tian-Tian Zhang, Huiying He, Chuan-Jia Xu, Qiantang Fu, Yan-Bin Tao, Ronghua Xu, Zeng-Fu Xu

**Affiliations:** 1School of Life Sciences, University of Science and Technology of China, Hefei 230027, China; ttzz@mail.ustc.edu.cn; 2CAS Key Laboratory of Tropical Plant Resources and Sustainable Use, Xishuangbanna Tropical Botanical Garden, The Innovative Academy of Seed Design, Chinese Academy of Sciences, Menglun, Mengla 666303, China; hhy@xtbg.org.cn (H.H.); xuchuanjia@xtbg.ac.cn (C.-J.X.); fuqiantang@xtbg.ac.cn (Q.F.); 3Center of Economic Botany, Core Botanical Gardens, Chinese Academy of Sciences, Menglun, Mengla 666303, China; 4College of Life and Health Sciences, Anhui Science and Technology University, Fengyang 233100, China; 5State Key Laboratory for Conservation and Utilization of Subtropical Agro-Bioresources, College of Forestry, Guangxi University, Nanning 530004, China

**Keywords:** *J. curcas*, oil content, TAG, fatty acid, seed kernels, leaves

## Abstract

Diacylglycerol acyltransferase (DGAT) is the only enzyme that catalyzes the acyl-CoA-dependent acylation of sn-1, 2-diacylglycerol (DAG) to form triacylglycerol (TAG). The two main types of DGAT enzymes in the woody perennial biofuel plant *Jatropha curcas*, JcDGAT1 and JcDGAT2, were previously characterized only in heterologous systems. In this study, we investigated the functions of *JcDGAT1* and *JcDGAT2* in *J. curcas.*
*JcDGAT1* and *JcDGAT2* were found to be predominantly expressed during the late stages of *J. curcas* seed development, in which large amounts of oil accumulated. As expected, overexpression of *JcDGAT1* or *JcDGAT2* under the control of the *CaMV35S* promoter gave rise to an increase in seed kernel oil production, reaching a content of 53.7% and 55.7% of the seed kernel dry weight, respectively, which were respectively 25% and 29.6% higher than that of control plants. The increase in seed oil content was accompanied by decreases in the contents of protein and soluble sugars in the seeds. Simultaneously, there was a two- to four-fold higher leaf TAG content in transgenic plants than in control plants. Moreover, by analysis of the fatty acid (FA) profiles, we found that JcDGAT1 and JcDGAT2 had the same substrate specificity with preferences for C18:2 in seed TAGs, and C16:0, C18:0, and C18:1 in leaf TAGs. Therefore, our study confirms the important role of *JcDGAT1* and *JcDGAT2* in regulating oil production in *J. curcas*.

## 1. Introduction

*Jatropha curcas* L., belonging to the Euphorbiaceae family, is a perennial woody biofuel plant with great environmental adaptability [1,2]. The oil content of *J. curcas* seed kernels from different regions and altitudes ranges from 40% to 55% [3,4,5], and its major fatty acids (FAs) are palmitic acid (14%), stearic acid (7%), oleic acid (44%) and linoleic acid (32%) [6]. Such profiles of kernel oil with high amounts of monounsaturated and polyunsaturated FAs determine its semi-drying property and make it suitable as an efficient substitute for diesel fuel [2,6]. However, unsatisfied seed and oil yields and low economic returns limit the application of *J*. *curcas* in the biodiesel industry [7,8,9]. Therefore, breeding for high *J. curcas* oil yield by genetic improvement has great potential for future biodiesel production.

Triacylglycerols (TAGs), which are composed of three FA chains esterified to a glycerol backbone, are the major component of plant seed oil [10]. The formation of TAGs generally consists of two processes: FA synthesis in plastids and TAG assembly in the endoplasmic reticulum [11]. Diacylglycerol acyltransferase (DGAT), which is considered to be a crucial enzyme for TAG biosynthesis, catalyzes the acylation of the *sn*-3 position of DAG to form TAG in the last step in the Kennedy pathway [12]. DGAT1 and DGAT2 are two main types of DGAT enzymes identified in eukaryotes [13]. Although both of these enzymes are endoplasmic reticulum membrane-bound enzymes, evolutionary analyses demonstrated that DGAT1 and DGAT2 evolved separately and belonged to unrelated families with respect to protein sequence and structure [13,14]. For instance, DGAT1 contains 8–10 transmembrane domains, whereas DGAT2 contains only two such domains [12,13,15].

*DGAT1* and *DGAT2*, which encode the two types of DGAT, are probably the most extensively studied genes for modifying oil production in plant species [12,16,17]. The expression levels of *DGAT1* and *DGAT2* were normally correlated with TAG accumulation. In *Arabidopsis*, *AtDGAT1* rather than *AtDGAT2* contributes to TAG accumulation in seeds, which was revealed by upregulation and mutational downregulation studies [18,19]. However, transient expression of *AtDGAT2* in tobacco leaves gave rise to an increase in the TAG content to levels twice as high as that resulting from the parallel expression of *AtDGAT1* [20]. *AtDGAT1* might have effects that favor TAG accumulation in seeds, whereas *AtDGAT2* may be more important for TAG accumulation in vegetative tissues. Many additional studies reported that *DGAT1* plays an important role in oil production [21,22,23,24,25]. However, *GmDGAT2D* from soybean (*Glycine max*) and *PtDGAT2B* from *Phaeodactylum tricornutum* could increase the oil content, as *GmDGAT1* and *PtDGAT1* did [26,27,28,29]. Particularly, in *Cyperus esculentus*, whose tuber is a lipid storage organ, it seems that only *CeDGAT2b* contributes to TAG accumulation because *CeDGAT1* and *CeDGAT2a* failed to restore the TAG-deficient phenotypes of yeast and *Arabidopsis* and increase the TAG content in wild-type *Arabidopsis* [30].

In addition to the modification of oil content, *DGAT1* and *DGAT2* can affect the FA compositions of TAGs with different substrate preferences. DGAT1 incorporates the usual FAs such as palmitic acid, stearic acid, oleic acid and linoleic acid into TAGs, whereas DGAT2 prefers unusual FAs. For example, *Ricinus communis RcDGAT2* successfully increased the amount of hydroxy FA (HFA) that constitutes ricinoleic acid in *Arabidopsis*, whereas *RcDGAT1* failed [31]. Similarly, *DGAT2* from the tung tree (*Vernicia fordii*) rather than *DGAT1* could give rise to a substantial increase in eleostearic acid, the major component of tung oil, in yeast and *Arabidopsis* leaves [15,32]. *VgDGAT2* from *Vernonia galamensis* had a much greater impact on vernolic acid accumulation in soybean seeds than *VgDGAT1* [33]. Therefore, *DGAT2* plays an essential role in unusual TAG synthesis.

*JcDGAT1* and *JcDGAT2* from *J. curcas* have been characterized in yeast, tobacco, and *Arabidopsis* systems, demonstrating that they function effectively in TAG biosynthesis [34,35]. In this study, *JcDGAT1* and *JcDGAT2* were overexpressed in *J. curcas* under the control of the *CaMV 35S* promoter to enhance oil production. The results showed that the total oil production in the seed kernels and leaves was significantly increased in the *JcDGAT1*- and *JcDGAT2*-overexpressing *J. curcas* plants. However, in terms of FA profiles, different alterations occurred between the seed kernels and leaves of *JcDGATs*-overexpressing *J. curcas*, and no difference between the two types of transgenic plants was found.

## 2. Results

### 2.1. JcDGAT1 and JcDGAT2 are Highly Expressed at the Late Stages of Seed Development

To analyze the expression patterns of *JcDGAT1* and *JcDGAT2* in wild-type *J. curcas*, quantitative real-time PCR (qRT-PCR) was performed using total RNA extracted from various tissues from adult *J. curcas,* including roots, stems, young and mature leaves, female and male flowers, green pericarps, and seeds at different developmental stages in which seeds matured at 49 days after pollination (DAP). As shown in Figure 1, both *JcDGAT1* and *JcDGAT2* were highly expressed in the seeds at 42 and 49 DAP and constitutively expressed in other tissues at much lower levels. The results indicate that both *JcDGAT*s may be important for seed oil accumulation because the oil content increases remarkably at the late stages during seed development [3].

### 2.2. Overexpression of JcDGAT1 and JcDGAT2 Enhanced Seed Oil Production in Transgenic J. curcas

To understand whether the two *JcDGATs* can affect native oil production, both genes driven by the *CaMV35S* promoter were transformed into *J. curcas*. The obtained transgenic plants did not exhibit any morphological changes compared with the control plants harboring a construct of *β-glucuronidase* (*GUS*) driven by the constitutive *JcUEP* promoter (*JcUEP:GUS*) [36]. In total, we generated 19 independent *35S:JcDGAT1* and 18 independent *35S:JcDGAT2* transgenic *J. curcas* lines, which were identified by the PCR detection of *35S:JcDGAT*s fragments (Appendix A). After *J. curcas* plants were fully mature, the T1 seeds were harvested and appeared normal relative to the control seeds (Figure 2A). The expression levels of *JcDGAT1* and *JcDGAT2* were significantly increased in the transgenic seeds (Figure 2B). Then, we detected the seed kernel oil contents of five independent transgenic lines from each transgenic type. Compared with control plants, the total kernel oil contents in all tested transgenic lines were increased (Figure 3A), with no significant difference in the seed dry weight (Figure 3B). The *35S:JcDGAT1* and *35S:JcDGAT2* transgenic lines accumulated 13.8% to 25% and 18.9% to 29.6% more kernel oil, respectively. This result indicated that both *JcDGAT1* and *JcDGAT2* were able to promote seed oil production in *J. curcas*. Additionally, we found that the increase in seed oil contents were positively correlated with the expression levels of *JcDGAT1* and *JcDGAT2*. L36 and L40 from *35S:JcDGAT1* transgenic lines and L51 and L31 from *35S:JcDGAT2* transgenic lines, which exhibited intermediate and high expression levels of transgenes, respectively, were selected for further studies.

In addition, we examined the protein, starch, and soluble sugar contents in the seeds from *35S:JcDGATs* transgenic lines to determine possible changes in other metabolites along with the increased oil content. The results showed that the increase in seed oil content was primarily compensated by decreases in the protein and soluble sugar contents in both transgenic events (Table 1). The starch content in transgenic lines was not significantly different from that in the control, except for the decrease in the *35S:JcDGAT1* transgenic line L40. The greater the increase in the oil content was, the more the contents of other metabolites decreased.

### 2.3. FA Compositions Significantly Changed in Seed Oil of Transgenic J. curcas

Since the oil contents in transgenic lines were increased, we assumed that the FA compositions would also be changed. Therefore, four major FAs in *J. curcas* seed oil, palmitic acid (C16:0), stearic acid (C18:0), oleic acid (C18:1) and linoleic acid (C18:2) [37], were detected by gas chromatography flame ionization detection (GC-FID). The results showed a shift from saturated FAs (SFAs, C16:0 and C18:0) and monounsaturated FAs (MUFAs, C18:1) towards polyunsaturated FAs (PUFAs, C18:2) in the transgenic lines when compared with control plants (Figure 4). In *35S:JcDGAT1* transgenic lines, the contents of C16:0, C18:0 and C18:1 were reduced on average by 16%, 18% and 10%, respectively, but that of C18:2 was increased on average by 33%. In *35S:JcDGAT2* transgenic lines, the C16:0, C18:0 and C18:1 contents were reduced on average by 9%, 49% and 17%, whereas that of C18:2 was increased on average by 46%. The results reveal that JcDGAT1 and JcDGAT2 had a similar effect on the FA profiles in *J. curcas* seed oil with a preference of using C18:2 as the acyl donor. In summary, overexpression of *JcDGAT*s resulted in a decrease in the saturation level and an increase in the unsaturation level.

### 2.4. Overexpression of JcDGAT1 and JcDGAT2 Altered TAG Accumulation and FA Compositions in Transgenic J. curcas Leaves

Furthermore, we investigated the effect of *JcDGAT*s on TAG synthesis in leaves. Total lipids extracted from the fully expanded green leaves (Figure 5A), in which the expression levels of transgenes were also significantly increased (Figure 5B), were analyzed for the separation of TAGs using thin-layer chromatography (TLC) (Appendix A). After TAG quantification by GC-FID, we found that the TAG contents in leaves of transgenic lines were higher than those in control plants (Figure 5C). The highest TAG content of 170 µg/100 mg dry weight detected in the leaves from the *35S:JcDGAT1* transgenic line L36 was almost four times that of the control (43 µg/100 mg), which was consistent with the TLC results showing that L36 exhibited the most intense TAG spot (Appendix A). The TAG contents in the other transgenic lines increased by an average of two times compared to that of the control.

In addition, compared with the control, the FA compositions of TAGs in transgenic plants were significantly altered (Figure 6). The contents of C16:0, C18:0 and C18:1 in all transgenic lines were markedly increased except in the *35S:JcDGAT2* transgenic line L51. The alterations in these three FAs were contrary to those in seed oil. The C18:2 contents in *35S:JcDGAT1* transgenic line L40 and *35S:JcDGAT2* transgenic line L51 were also increased. These increased FAs were at the expense of C16:1, C18:3 and other FAs. The results indicate that JcDGAT1 and JcDGAT2 preferred to use C16:0, C18:0 and C18:1 as acyl donors in leaf TAG synthesis.

## 3. Discussion

DGAT, which catalyzes the last step in the Kennedy pathway, is especially important for TAG accumulation. Although there are at least four types of DGATs [38,39], type 1 and type 2 DGATs are reported to contribute to TAG synthesis in most plant species. In *J. curcas*, three DGAT genes, *JcDGAT1*, *JcDGAT2*, and *JcDGAT3*, have been identified [37,40], in which *JcDGAT1* and *JcDGAT2* have been proven to affect oil accumulation in *Arabidopsis* and tobacco [34,35]. In addition, virus-induced gene silencing of a transcription factor *JcMYB1* in *J. curcas* leaves down-regulated the expression of *JcDGAT1*, resulting in a reduction in the lipid content [41]. In the current study, *JcDGAT1* and *JcDGAT2* were overexpressed in *J. curcas* to improve the oil yield. Oil contents were increased by approximately 25% and 29.6% in seed kernels (Figure 3A and Table 1) and four- and two-fold in leaves (Figure 5C) of *JcDGAT1-* and *JcDGAT2-*overexpressing *J. curcas*, respectively. Our results demonstrate that both *JcDGAT1* and *JcDGAT2* play an effective role in lipid accumulation and perform similar functions in seeds and leaves. Our conclusions further support the results from the heterologous overexpression of *JcDGAT1* and *JcDGAT2* in yeast and tobacco by Xu et al. [34]. Similar to the functions of *JcDGAT1* and *JcDGAT2*, *GmDGAT1A* and *GmDGAT2D* from soybeans can both promote TAG accumulation in soybean hairy roots [28]. The similar role in oil accumulation of the two types of *DGAT* genes from *J. curcas* and soybean is associated with their similar expression patterns. Transcripts of *GmDGAT1A* and *GmDGAT2D* were both found in soybean roots [27,28]. In *J. curcas*, *JcDGAT1* and *JcDGAT2* were both expressed in the seeds and leaves (Figure 1). Moreover, the increases in expression levels of *JcDGAT1* and *JcDGAT2* in transgenic *J. curcas* were higher in leaves (Figure 5B) than in seeds (Figure 2B), which indicated the *35S* promoter was more active in mature leaves than in mature seeds. Our results show that the kernel oil content increased along with the reduction in the contents of other major metabolites to varying degrees and did so significantly in *35S:JcDGAT1* transgenic line L40 and *35S:JcDGAT2* transgenic line L31, in which the protein contents decreased by 18% and 17% and the soluble sugar contents decreased by 31% and 11%, respectively, compared with the contents in the control lines (Table 1). During seed maturation, carbon and nitrogen are present in three main forms: TAGs, carbohydrates, and proteins [42]. Many studies have shown that the flow of carbon determines the accumulation of these three types of compounds [43,44]. Sugar produces pyruvate, the precursor of acetyl-CoA, through glycolysis, and pyruvate plays a central role in the production of TAGs, starch, and amino acids. The greater carbon flux into the lipid metabolism pathway in *JcDGATs*-overexpressing plants resulted in higher TAG contents, which must be accompanied by a reduction in protein and sugar contents. It has been shown that DGAT not only incorporates the substrate into TAGs but also increases the flow of products from the upstream glycolytic pathway to lipid synthesis [45]. Therefore, this indicates that the accumulation of oil can be increased by enhancing de novo FA synthesis, such as through the overexpression of *WRI1*, which can regulate the allocation of carbon between carbohydrates and storage lipids, or by the prevention of starch or protein formation from pyruvate, such as downregulation of *ADP-glucose pyrophosphorylase*, whichis a rate-limiting enzyme for starch synthesis [46,47].

Our data also indicate that JcDGAT1 and JcDGAT2 have similar effects on the FA profiles of seed oil. In both *JcDGAT1-* and *JcDGAT2-*overexpressing lines, the percentage of C18:2 significantly increased at the expense of C16:0, C18:0 and C18:1 (Figure 4), suggesting that both JcDGAT1 and JcDGAT2 have a substrate preference for linoleic acid. In *JcDGATs*-overexpressing tobacco, however, *JcDGAT1* did not induce changes in FA compositions, whereas *JcDGAT2* overexpression led to a significant increase in the proportion of C18:2 [34]. In *Arabidopsis* seeds, the expression of *JcDGAT1* did not affect SFAs but increased the percentages of C18:3 and C18:2 largely at the expense of C18:1 [35]. The variation in FA composition induced by *DGAT* genes may be caused by different substrates and selectivity in distinct species. For example, overexpression of *Sesamum indicum SiDGAT1* in *Arabidopsis* and soybean seeds also showed different changes in FA compositions [22]. In addition, we noted a significant increase in the proportion of UFAs and a reduction in SFAs in *JcDGATs* transgenic seeds (Figure 4). The degree of (un)saturation of kernel oil is important for determining biodiesel oxidative stability and performance properties [48]. A high level of polyunsaturated FAs, such as linoleic acid, can reduce the stability of biodiesel due to its susceptibility to oxidation and is not ideal for biodiesel applications [48]. However, vegetable oils rich in oleic acid, which contain one double bond, can be better used as raw materials for biodiesel, and oil that is high in oleic acid and low in SFAs can improve the oxidation stability while increasing the cold flow performance [49,50,51]. Therefore, based on our results, we expect to further improve the quality of *J. curcas* seed oil by increasing the proportion of oleic acid. Previous reports showed that the percentage of C18:1 was increased to 50–60% in *JcFAD2-1* RNAi transgenic *J. curcas* endosperm [52]. Hence, we speculate that silencing the *JcFAD2* gene in *JcDGATs-*overexpressing *J. curcas* may result in an improvement in both the quality and yield of seed oil.

Moreover, overexpression of *JcDGA1* and *JcDGAT2* in *J. curcas* leaves caused significant alterations in the FA profiles of TAGs. Both JcDGAT1 and JcDGAT2 preferred to incorporate the C16:0, C18:0 and C18:1 FAs but excluded C18:3 and C16:1 into TAGs in transgenic leaves (Figure 6). Such changes in C18:3 conversion to C18:1 were also observed in *J. curcas* leaves with overexpression of *AtDGAT1* [53]. However, compared to *AtDGAT1*, *JcDGATs* had a better effect on the production of C18:1 in the leaves of *J*. curcas (increased by 1.3 to 6-fold compared with 20–31%). The alterations of FA compositions in the TAGs of *JcDGATs-*overexpressing transgenic leaves seem to be favorable for biodiesel feedstock. C16:1 was enriched in the phosphatidylglycerol (PG) component of the photosynthetic membranes [54]. Our results show that C16:1 was significantly reduced, which seems to indicate that JcDGAT1 and JcDGAT2 are also active in the chloroplast membrane at the leaves’ maturity [55] but play a role in excluding C16:1 from TAGs. Interestingly, we also observed that the changes in FA compositions in leaves were distinct from those in seed kernels. The C16:0, C18:0 and C18:1 contents increased in transgenic leaves (Figure 6) but decreased in transgenic seed kernels (Figure 4). This may be because the profiles of FAs in TAGs are determined not only by the substrate preferences of the DGATs but also by the availability of substrate pools, including acyl-CoA and DAG [12]. In most seed oils, the phosphatidylcholine (PC) backbones carrying PUFAs can be catalyzed by phospholipase D (PLD), phospholipase C (PLC) or phosphatidylcholine:diacylglycerol cholinephosphotransferase (PDCT) to form phosphatidic acid (PA) or DAG. These PC-derived PA or DAG will enter the Kennedy pathway to form TAG in the last step catalyzed by DGAT [44,56]. However, in *J. curcas* seeds, a sharp downregulation of the expression of a linoleate desaturase (*FAD3*) gene, which functions in the PC pool, resulted in a reduction in the biosynthesis of linolenic acid at maturity [3]. In *J. curcas* leaves, however, we found that the proportion of linolenic acid in total TAGs was as high as 29% in the control plants (Figure 6), indicating that the substrate pool of DGATs was different between seeds and leaves in *J. curcas*. In addition, a previous study suggested that leaves may have an inherent capacity for TAG synthesis compared with seeds [57]. For example, in the mutant AS11 of *Arabidopsis*, which affected DGAT activity, the FA profiles were significantly altered in seeds but not in leaves [58]. Thus, DGATs may have distinct effects on lipids in different tissues or different species.

Overall, we increased the oil contents of *J. curcas* seeds and leaves by overexpressing *JcDGAT1* and *JcDGAT2*. This is the first study to use two types of *JcDGATs* to improve oil production in *J. curcas*. However, we need to further improve the yield and quality of oil in *J. curcas* to meet requirements for biodiesel feedstocks. Coexpressing *JcDGATs* with an antisense inhibition of *JcFAD2* gene [52] in *J. curcas* could be a possibility for achieving better results.

## 4. Materials and Methods

### 4.1. Plant Materials

The adult *J. curcas* plants in this study were planted in Xishuangbanna Tropical Botanical Garden (XTBG; 21°54′ N, 101°46′ E; 580 m in altitude) of the Chinese Academy of Sciences, Yunnan, China [9].

### 4.2. Plant Transformation

The 35S:*JcDGAT1* and 35S:*JcDGAT2* vectors constructed by Xu et al. [34] and the *JcUEP:GUS* vector constructed by Tao et al. [36] were transferred to *Agrobacterium tumefaciens* EHA105 for *J. curcas* transformation. The method used for *J. curcas* transformation was described previously by Fu et al. [59]. The transgenic plants were identified by PCR detection of *35S:JcDGAT*s fragments and the primers are listed in Appendix A.

### 4.3. qRT-PCR Analysis

The expression levels of *JcDGAT*s in wild-type and transgenic *J. curcas* plants were analyzed by qRT-PCR. Total RNA was isolated [60] and reverse transcribed using the PrimeScript^®^ RT reagent kit (TAKARA Biotechnology, Dalian, China). qRT-PCR was performed using TB Green^®^ Premix Ex Taq™ II (TAKARA Biotechnology, Dalian, China) on a LightCycler 480II (Roche Diagnostics, Mannheim, Germany) device. All of the expression levels were normalized to the expression of *JcActin1* [61]. The primers used in the qRT-PCR assay are listed in Appendix A.

### 4.4. Lipid Analysis

Seed oil was extracted as described previously [62]. Mature seeds were harvested from adult *J. curcas* and dried to a constant weight. Ten seeds were weighed as a biological replicate. Seeds were removed from their shells and ground into a powder mixture. Oil was extracted from 100 mg of the mixtures with hexane three times. Subsequently, the collected hexanes portions were evaporated at 42 °C. The oil content of the kernels is presented as the percentage of dry weight of the kernels. This experiment was replicated three times. Since the TAG content can reach up to 97% of the oil in mature seeds [63], the total oil extracted from mature kernels was considered the TAG content in this study.

Approximately 10 mg of the extracted oil shown above was methylated in 2% methanol-H_2_SO_4_. Then, fatty acid methyl esters (FAMEs) were separated and detected using GC-FID (PerkinElmer Clarus 680, Singapore) equipped with a 30 m × 0.25 μm × 0.32 mm (inner diameter) Elite-225 column (PerkinElmer, Singapore). The following temperature program was applied: 150 °C held for 3 min; 10 °C/min increase to 180 °C, held for 9 min; and 5 °C/min increase to 210 °C, held for 8 min [64]. The proportions of FAs were calculated via the peak area of each component compared to the total peak area [62].

### 4.5. TAG Analysis in Leaves

TAGs in leaves were analyzed using the method described in potato leaves [65]. Fully expanded green leaves were sampled and freeze-dried overnight. The dried leaves were divided into triplicate and weighed. Lipids were extracted in methanol/chloroform/0.1 M KCl (1:2:1, by volume) with a mixer mill (MM400, RETSCH Company, Germany) at 20 frequency/s for 3 min. After centrifugation, the bottom solution was collected in a glass tube and evaporated completely. Then, the lipids were dissolved in chloroform in 1 µl/mg DW. A total of 30 mgof lipids was taken for TLC on a plate (20 cm × 20 cm, silica gel 60, Merck) with a hexane/diethyl ether/acetic acid solvent system (70:30:1, by volume). After spraying with primuline (Sigma) in 80% aqueous acetone, the plate was visualized under UV. TAGs were separated by scraping the corresponding silica bands. Then, the TAG fraction was transmethylated and analyzed by GC-FID as described for seed lipid analysis. For TAG quantification, triheptadecanoin (C17:0-TAG) was added as an internal standard at 1 µg/mg DW. The TAG content was calculated by the formula: TAG content (% DW) = sum of peak area of the endogenous fatty acids/peak area of internal standard × concentration of internal standard (1 µg× 10^−3^/mg) %. The percentage of FAs in TAGs was calculated via the peak area of each component compared to the total peak area of endogenous FAs.

### 4.6. Quantifications of Protein, Starch, and Soluble Sugar

For the quantification of protein, 50 mg of the seed kernel mixture prepared as described in Section 4.4 was homogenized in 300 µl of extraction buffer containing 50 mM Tris-HCl (pH 7.5), 150 mM NaCl, 1 mM EDTA, and 0.1% Triton X-100 (*w*/*v*) [66]. Then, it was transferred to a 1.5-mL centrifuge tube and placed at 4 °C for 30 min. Following centrifugation at 1200 g for 10 min, the supernatant was collected and analyzed using a Bradford protein assay kit (BL524A, GUANGKE Technology Company, Kunming).

The contents of starch and soluble sugar seed kernels were examined using a Starch Assay Kit (A148-1-1, ZanNa Biological Company, Kunming) and Soluble Sugar Assay Kit (QYS-234076, QIYI Biological Technology Company, Shanghai), respectively.

## Figures and Tables

**Figure 1 plants-10-00699-f001:**
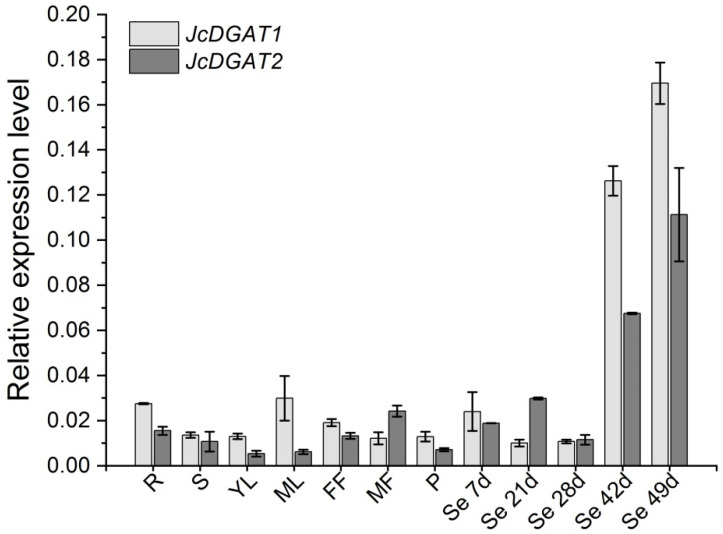
Expression patterns of *JcDGAT1* and *JcDGAT2* in *J. curcas*. The expression levels of *JcDGAT1* and *JcDGAT2* were analyzed in the roots (R), stems (S), young leaves (YL), mature leaves (ML), female flowers (FF), male flowers (MF), green pericarps (P), and seeds at 7, 21, 28, 42 and 49 days after pollination (DAP) (Se 7d, 21d, 28d, 42d and 49d, respectively). The qRT-PCR results were obtained from two independent biological replicates and three technical replicates. The levels were normalized using the amplified products of *JcActin1*. The values are presented as the means ± standard deviations.

**Figure 2 plants-10-00699-f002:**
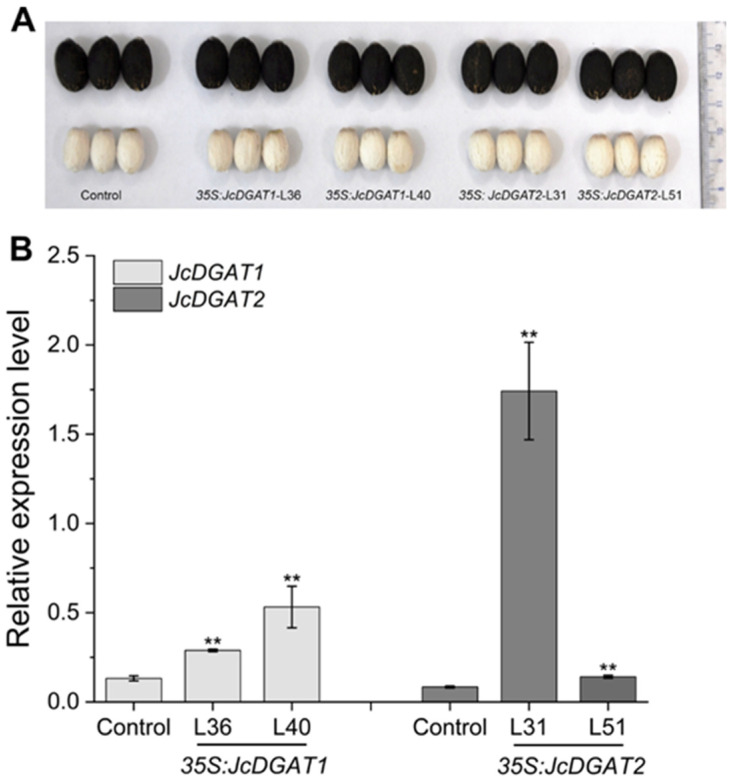
Appearance of seeds and kernels from *35S:JcDGATs* transgenic *J. curcas* lines. (**A**) Phenotypes of seeds and kernels from control and transgenic lines. (**B**) The expression levels of *JcDGAT1* and *JcDGAT2* in the control and *35S:JcDGATs* transgenic seed kernels. The expression levels were normalized using the amplified products of the *JcActin1*. The values are presented as the mean ± standard deviations (*n* = 3). The student’s t-test was used for statistical analyses. ** *p* ≤ 0.01.

**Figure 3 plants-10-00699-f003:**
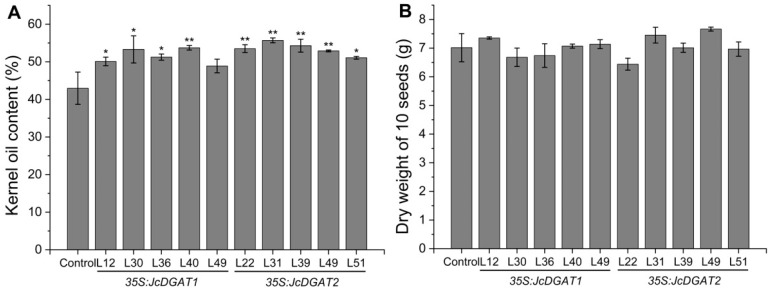
Seed kernel oil contents and dry weights in control and *35S:JcDGATs* transgenic lines. (**A**) Total kernel oil content. (**B**) Dry weight of 10 seeds. The values are presented as the means ± standard deviations of three independent biological replicates. The student’s t-test was used for statistical analyses. * *p* ≤ 0.05, ** *p* ≤ 0.01.

**Figure 4 plants-10-00699-f004:**
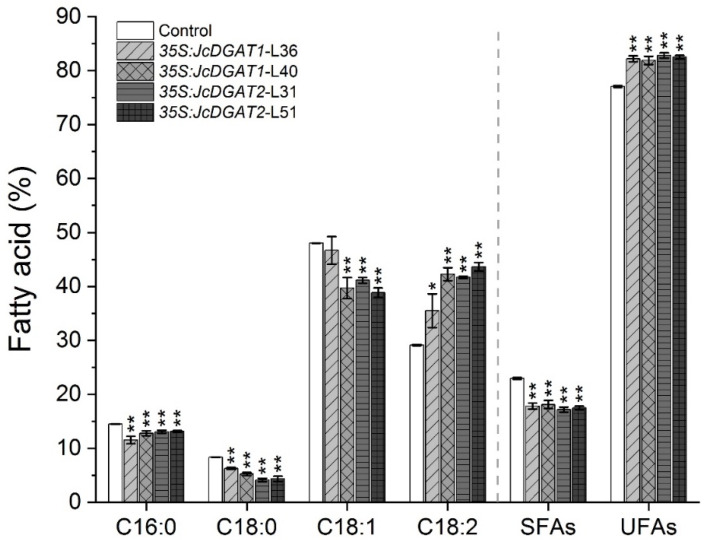
Fatty acid (FA) profiles of seed kernel oil in control and transgenic *J. curcas*. Saturated FAs (SFAs) indicate the sum of the contents of C16:0 and C18:0, and unsaturated FAs (UFAs) indicate the sum of the contents of C18:1 and C18:2. The values are presented as the mean ± standard deviation of three biological replicates. The student’s *t*-test was used for statistical analyses. * *p* ≤ 0.05, ** *p* ≤ 0.01.

**Figure 5 plants-10-00699-f005:**
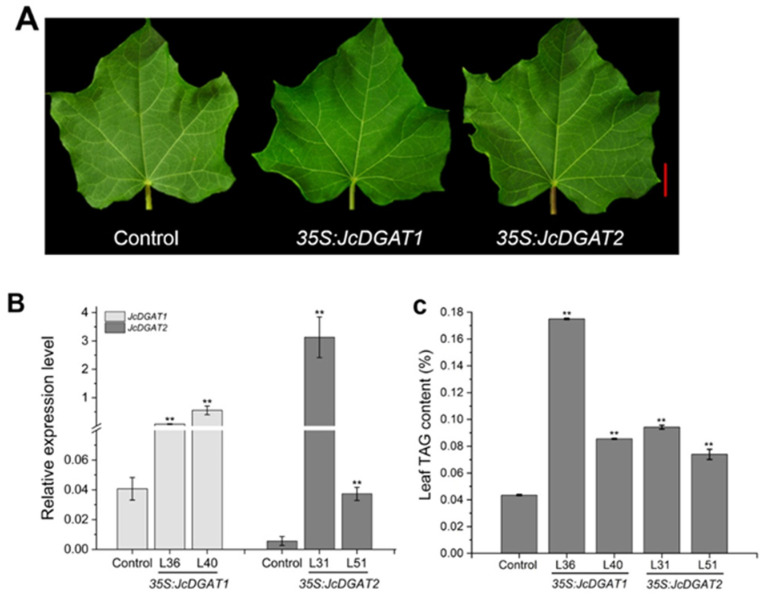
Triacylglycerol (TAG) contents accumulated in transgenic lines with normal phenotypes. (**A**) Phenotypes of leaves from the control and transgenic plants during the period of fruiting. Scale bar = 2 cm. (**B**) The expression levels of *JcDGAT1* and *JcDGAT2* in the control and *35S:JcDGATs* transgenic leaves. (**C**) TAG contents in the leaves of control and transgenic plants. The values are presented as the mean ± standard deviation of two biological replicates. The student’s *t*-test was used for statistical analyses. ** *p* ≤ 0.01.

**Figure 6 plants-10-00699-f006:**
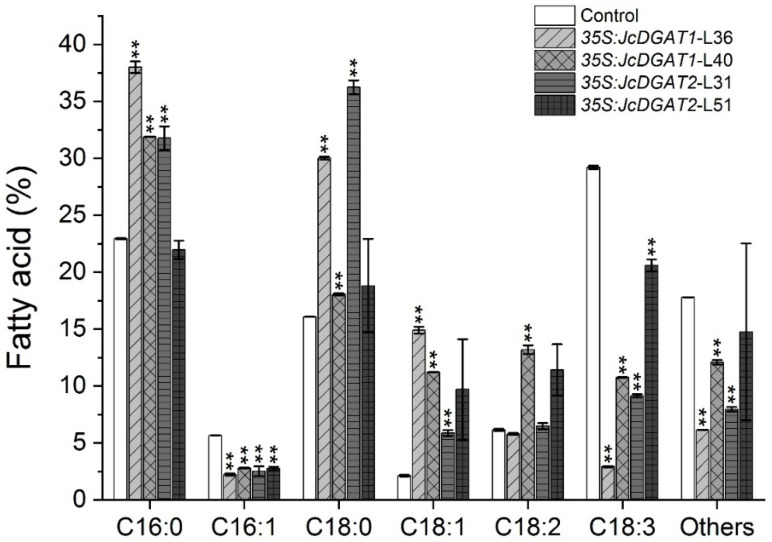
FA profile of TAGs in the leaves of control and transgenic plants. The values are presented as the mean ± standard deviation of two biological replicates. The student’s *t*-test was used for statistical analyses. ** *p* ≤ 0.01.

**Table 1 plants-10-00699-t001:** Content of major metabolites in the seed kernels from the control and *35S:JcDGATs* transgenic *J. curcas* plants.

Genotype	Oil (%, *w*/*w*)	Protein (%, *w*/*w*)	Starch (%, *w*/*w*)	Soluble Sugar (%, *w*/*w*)
Control	42.97 ± 4.29	19.15 ± 0.95	5.95 ± 0.93	3.15 ± 0.18
*35S:JcDGAT1*-L36	51.25 ± 0.82 *	17.24 ± 2.63	6.06 ± 1.20	2.96 ± 0.01
*35S:JcDGAT1*-L40	53.73 ± 0.61 **	15.61 ± 0.26 **	4.65 ± 0.52	2.17 ± 0.14 **
*35S:JcDGAT2*-L31	55.70 ± 0.64 **	15.83 ± 0.60 **	5.32 ± 0.93	2.79 ± 0.42
*35S:JcDGAT2*-L51	51.07 ± 0.37 *	17.66 ± 0.89	6.03 ± 1.01	2.88 ± 0.20

The values represent the means ± SD (*n* = 3). The student’s *t*-test was used for statistical analyses. * *p* ≤ 0.05, ** *p* ≤ 0.01.

## Data Availability

All data are available in this article and in the Appendix A.

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
