# Peer review of "Overexpression of Type 1 and 2 Diacylglycerol Acyltransferase Genes (JcDGAT1 and JcDGAT2) Enhances Oil Production in the Woody Perennial Biofuel Plant Jatropha curcas"

_plants, 2021, doi:10.3390/plants10040699_

Round 1
Reviewer 1 Report
This research analyzed the gene expression and oil content changes in the seeds and leaves of Jatropha curcas by overexpression of JcDGAT1 and JcDGAT2 genes. It is of some significance for us to understand the function of these two genes and improve oil production in the seeds, but the manuscript encountered some problems in experiment design and manuscript preparation.
- The authors should provide molecular evidences or published references to indicate that JcDGAT1 and JcDGAT2 genes were transformed into J. curcas. How many years for these transgenic plants to form seeds?
- Fig.1, the data from wild type (control) is missing, and the authors need to provide the data in this figure.
- Line 122 to 123, “The 35S:JcDGAT1 and 35S:JcDGAT2 transgenic lines accumulated 13.8 to 25% and 18.9 to 29.6% more kernel oil, respectively”, however, Fig.2 and Table 1 showed the difference between the percent values of transgenic plants and control is not more than 15%.
- The plant name was written as Jatropha in many sentences of the text, and the name should be Jatropha curcas or J. curcas.
Reviewer 2 Report
This manuscript describes a straight-forward and successful approach to modify the oil content of Jatropha curcas. Using over-expression of the native DGAT1 and DGAT2 ORFs under the control of a 35S promoter, the authors describe modified oil content and FA composition obtained using this approach in both seeds and leaves. This manuscript was very easy to read, and I just have a small number of comments.
1) How many total transgenic lines per construct were created? 50? The authors only characterize 2 lines per construct; why these 2? Were they the highest expressing transgenic lines obtained per construct? Were they randomly selected? Please provide some text explaining how many total lines were created, and why/how these lines were selected for further characterization.
2) Did the authors perform any molecular confirmation of transgene integration? Nothing is stated. Did they confirm these were single copy insertion lines? Please provide some comment in the text.
3) The authors provided an analysis of DGAT1/2 expression levels in wild-type Jatropha tissues/organs (Figure 1), with a nice timeline (7d, 21d, 28d, 42d, 49d) for seed development. I'm surprised that they did not provide an expression analysis for the transgenic lines during seed development, as they did for the leaf (Figure S1B). Given the 35S promoter used with the transgenics, I would have liked to see a comparison of DGAT1 and 2 expression during the same seed developmental time course to assess temporal differences in expression relative to WT, and also to compare overall expression levels in the transgenic lines (driven by endogenous promoter + 35S promoter) compared with WT (endogenous promoter only). We know that the seed oil contents were increased for DGAT1 (up to 25%) and DGAT2 (up to 29.6%) lines; did this equate with similar overall increases in transcript abundance, or was transcript abundance much higher/lower? Did the 35S promoter drive an earlier accumulation of oil? An analysis of gene expression in the transgenic lines would contribute to an understanding as to what is happening. Can the authors add a seed expression analysis figure for the transgenic lines?
4) The overall promotion of oil accumulation in transgenics over WT levels was much greater in the leaf than in the seed, and I'm assuming this may be due to the strength of the 35S promoter over the endogenous DGAT1/2 promoters in the seed vs leaf. Thoughts or comments on this that could be added to discussion?
5) A 2019 paper [Khan et al. 2019. JcMYB1, a Jatropha R2R3MYB transcription factor gene, modulates lipid biosynthesis in transgenic plants. Plant and Cell Physiology 60(2):462-475.] looked at an R2R3 MYB from Jatropha that binds to the Jatropha DGAT1 promoter, and the impact of its OE and knockdown on oil accumulation and FA content. This paper is another approach to modify Jatropha oil content via manipulation of DGAT1 expression and could be included in their discussion.
Reviewer 3 Report
The present work “Overexpression of type 1 and 2 diacylglycerol acyltransferase genes (JcDGAT1 and JcDGAT2) enhances oil production in the woody perennial biofuel plant Jatropha curcas” by Zhang et al. investigated the functions of JcDGAT1 and JcDGAT2 in J. curcas by overexpression. The result obtained showed an increase in seed kernel oil production, seed kernel dry weight compared with control plants.
Although the work may have important implications for this crop, I do not think that in the present form it can be suitable for Plants Journal. Furthermore, the whole manuscript requires extensive correction of the English.
e.g in the abstract “heterogenous systems”
Round 2
Reviewer 1 Report
The authors have revised the manuscript.
Reviewer 3 Report
Dear authors, the paper has been improved after reviewing.
Now is worthy of publication in the present form
This manuscript is a resubmission of an earlier submission. The following is a list of the peer review reports and author responses from that submission.